# A Novel Resorbable Composite Material Containing Poly(ester-co-urethane) and Precipitated Calcium Carbonate Spherulites for Bone Augmentation—Development and Preclinical Pilot Trials

**DOI:** 10.3390/molecules26010102

**Published:** 2020-12-28

**Authors:** Claudia Rode, Ralf Wyrwa, Juergen Weisser, Matthias Schnabelrauch, Marijan Vučak, Stefanie Grom, Frank Reinauer, Adrian Stetter, Karl Andreas Schlegel, Rainer Lutz

**Affiliations:** 1Biomaterials Department, INNOVENT e. V., Prüssingstrasse 27B, 07745 Jena, Germany; cr@innovent-jena.de (C.R.); rw1@innovent-jena.de (R.W.); jw1@innovent-jena.de (J.W.); 2Schaefer Kalk GmbH & Co. KG, Louise-Seher-Straße 6, 65582 Diez, Germany; marijan.vucak@schaeferkalk.de; 3Karl Leibinger Medizintechnik GmbH & Co. KG, a Company of the KLS Martin Group, Kolbinger Straße 10, 78570 Mühlheim an der Donau, Germany; stefanie.Grom@klsmartin.com (S.G.); F.Reinauer@klsmartin.com (F.R.); 4Clinic for Oral and Maxillofacial Surgery, Universitätsklinikum Erlangen, Glückstrasse 11, 91054 Erlangen, Germany; adrian.stetter@me.com (A.S.); andreas.schlegel@prof-schlegel.de (K.A.S.); rainer.lutz@uk-erlangen.de (R.L.)

**Keywords:** bioresorbable composite, calcium carbonate, degradable polyurethane, foam, bone regeneration

## Abstract

Polyurethanes have the potential to impart cell-relevant properties like excellent biocompatibility, high and interconnecting porosity and controlled degradability into biomaterials in a relatively simple way. In this context, a biodegradable composite material made of an isocyanate-terminated co-oligoester prepolymer and precipitated calcium carbonated spherulites (up to 60% *w*/*w*) was synthesized and investigated with regard to an application as bone substitute in dental and orthodontic application. After foaming the composite material, a predominantly interconnecting porous structure is obtained, which can be easily machined. The compressive strength of the foamed composites increases with raising calcium carbonate content and decreasing calcium carbonate particle size. When stored in an aqueous medium, there is a decrease in pressure stability of the composite, but this decrease is smaller the higher the proportion of the calcium carbonate component is. In vitro cytocompatibility studies of the foamed composites on MC3T3-E1 pre-osteoblasts revealed an excellent cytocompatibility. The in vitro degradation behaviour of foamed composite is characterised by a continuous loss of mass, which is slower with higher calcium carbonate contents. In a first pre-clinical pilot trial the foamed composite bone substitute material (fcm) was successfully evaluated in a model of vertical augmentation in an established animal model on the calvaria and on the lateral mandible of pigs.

## 1. Introduction

During the last decade, new concepts have been developed to generate biocompatible and biodegradable polyurethane (PU) and PU composite materials [1,2,3,4,5] with a great variability in their mechanical behaviour to fulfill the basic requirements of different types of tissues. At the same time, special attention has been paid to provide materials with adaptable degradation profiles releasing non-toxic degradation products [6,7].

Classical polyurethanes are generated from three major components: aliphatic or aromatic di- or tri-isocyanates, an oligomeric di- or polyol and a low molecular weight chain extender, mainly a diol or diamine (in case of polyurea formation) to increase the molecular weight [8]. The reaction of these three main components result in the formation of segmented polymers containing alternating soft segments formed by the di-or polyol and hard segments generated by the reaction of the isocyanates with the chain extender [9,10]. By using polyfunctional components branched or cross-crosslinked PU materials can be obtained [6,11,12]. Both the chemical nature of the starting components and their proportions determine the final set of PU material properties [13,14,15]. Another promising approach for adapting specific application properties offers the development of composite materials of PUs and inorganic fillers [16,17,18,19,20]. Concerning biodegradable PUs the formation of toxic degradation products arising from aromatic isocyanates is a major drawback for medical applications [21,22,23]. Therefore, in the design of biodegradable PUs for medical applications aliphatic isocyanates like 1,6-hexamethylene or 1,4-butane diisocyanate are often used instead of rather more reactive aromatic ones for the synthesis of PUs [1,23].

Recently, isocyanate building blocks derived from natural products as fatty acids [24], or natural amino acids [6] have been reported. Lysine derived isocyanates like L-lysine di- or triisocyanate esters are commercially available and lead to biodegradable PUs forming nontoxic degradation products [6,25,26,27,28,29,30,31].

Although urethane bonds can be cleaved hydrolytically under the influence of the body fluid medium and in the presence of enzymes [32,33,34], the introduction of more hydrolytically labile segments into the polymer backbone is commonly necessary to receive degradation levels relevant for tissue regeneration purposes. Both PUs and polyureas with faster rate of degradation can be synthesized by introducing soft segments with hydrolytically labile structures like poly(α-hydroxyester)s or their copolymers [35,36,37]. It is known that the degradation of these polyester materials in vivo is a bulk degradation process leading to the formation of acidic degradation products [38,39]. Since the process is of autocatalytic nature, fast degradation may cause problems in implant surrounding tissue sometimes. Therefore, pH compensation is suggested by composite formation with inorganic fillers similar to the mineral bone phase [40]. In this context calcium phosphates like hydroxyapatite or tri-calcium phosphate have been widely used in bone implant applications as filler material [41,42,43,44,45], however its buffering effect is rather marginal due to its insolubility. In contrast, calcium carbonate (CaCO_3_) has been recognized as inorganic bone filling material with a high buffering capacity and its osteoconductivity was approved in several studies [46,47,48].

Another important issue is the reaction of isocyanate-terminated prepolymers with water forming carbamic acid which decomposes to amine and carbon dioxide [49,50,51]. By reaction of the formed amine with excess isocyanates a 3D network is build up whereas the carbon dioxide acts as a foaming agent. The resulting porous material structures are advantageously for cell ingrowth and should improve the incorporation of implants in the surrounding tissue [52]. In this context it should be mentioned that classical PU’s are currently used as artificial bone imitation material for mechanical testing of orthopaedic devices [53].

The objective of the present study was to develop a biodegradable composite made of PU foams with precipitated calcium carbonate spherulites and the in vitro and in vivo evaluation of the composite as bone substitute in dental augmentation. 

It was our hypothesis that in this way a composite system can be provided which represents a both biodegradable and highly cytocompatible biomaterial for use in bone regeneration, and is further characterised by an open-pore structure to facilitate implant integration into the bony tissue. By using an inorganic filler with a sufficient buffer capacity, the known disadvantage of a pH decrease at the bony implant site during polymer degradation should be avoided at the same time. 

## 2. Results

### 2.1. Prepolymer Synthesis and Characterisation

The prepolymer was synthesized in a two-step procedure [30]. After preparing the PLA-*co*-PCL oligoester **1a** by ring-opening-polymerization of l-lactide and ε-caprolactone in the presence of meso-erythritol as initiator, the latter was treated with LDI to introduce isocyanate groups onto the terminal hydroxyl groups of the oligolactone giving the isocyanate terminated prepolymer ITP **1b**. The syntheses are depicted in Scheme 1 and the characteristics of the prepolymers **1a** and **1b** are found in Table 1.

The chemical structure of the 4-arm PLA-*co*-PCL oligoester **1a** and its isocyanate terminated prepolymer **1b** was confirmed by ^1^H-NMR spectroscopy. The molecular weights (M_n_, M_w_), were determined by GPC measurement. Especially the measured value for **1a** for **1a** (M_n_ = 2350 g/mol) shows close agreement with the calculated molecular weight (2354 g/mol) based on the used molar ratio of the starting components.

### 2.2. Preparation of Calcium Carbonate-Containing fCM

For foaming prepolymer **1b**, an aqueous DABCO solution was used to accelerate the rate of reaction as previously described [41]. Under these conditions, foaming times of about 15 min could be achieved. Clinically more optimal foaming times of about 4 min can be obtained by addition of small amounts of both LDI and DMSO (please see experimental part for the optimal foaming procedure).

The addition of CaCO_3_ spherulites leads to further acceleration of foaming, dependent on the used CaCO_3_ amount. As an example, foaming starts after 3.5 min if 30% CaCO_3_ (*w*/*w*) is added and after 3 min in case of 40%. With an increasing CaCO_3_ amount the pores of the fCM are decreasing accompanied by a rising density (Figure 1). The measurements of the density and the open/closed cell volume of fCM with different amounts of CaCO_3_ (SP01) are depicted in Figure 2. With rising CaCO_3_ content an increasing density can be observed and at the same time the open cell volume is decreasing.

### 2.3. Compressive Properties

At first the compressive strength was measured as a function of the CaCO_3_ particle size Figure 3a. A CaCO_3_ content of 60% (*w*/*w*) and 80 µL/g DMSO as softener were used. It was found that the compressive strength of the composite foams decreased with increasing particle size of the CaCO_3_ filler. Thus, with the CaCO_3_ particle size of 5 µm (SP01, brown curve) the highest compressive strengths of about 0.89 MPa were found, whereas specimens with a particle size of 12 µm (type 47, green curve) had compressive strengths of 0.39 MPa followed by the lowest values of 0.31 MPa for CaCO_3_ particle size of type 02 (diameter: 22 µm, purple curve). Next, the compressive strength was investigated varying the content of CaCO_3_ (SP01) in the fCM from 30% up to 60% maintaining a constant content of 80 µL/g DMSO as softener. As shown in Figure 3b, the compressive strength increased with increasing CaCO_3_ content in the composites from 0.12 to about 0.89 MPa. Furthermore, the compressive strength was influenced by the content of DMSO accelerating the foaming rate and acting as a softener. Figure 3c shows that the compressing strength was substantially decreasing with increasing DMSO amounts. Without adding DMSO the compression strength reached 19.9 MPa but decreased to 0.89 MPa at a DMSO content of 80 µL/g.

The compressive strength of fCM specimens containing 60% CaCO_3_ (SP01, *w*/*w*) was initially measured in a dry state and then after storage in PBS medium at 37 °C at intervals of 7 days (Figure 4). It has been observed that the compressive strength decreased after contact with liquid medium from about 19.8 MPa in the dry state to about 1.7 MPa after 2 h of storage in PBS. The higher the CaCO_3_ content in the composites, the higher was the compressive strength. This trend extended over the entire period of storage in PBS. After 119 days, specimens without CaCO_3_ started to dissolve and were discarded. After 154 days, samples with CaCO_3_ started to decompose.

### 2.4. Hydrolytic Degradation of Composite Foams

The in vitro degradation of fCM with varying CaCO_3_ contents between 0% and 60% was monitored determining the pH-value and weight loss of samples incubated in PBS or water, respectively. Every 28 days the weight change was determined gravimetrically.

It was found that specimens without CaCO_3_ released acidic components lowering the pH to values between 3 and 4 if stored in water but only to about 6.5 if stored in PBS. The pH values of specimens with added CaCO_3_ were in the range between 6.8 and 8.2 when stored in water. Higher CaCO_3_ contents resulted in higher pH-values. That means that CaCO_3_ in the composite foams was able to act as an effective buffer for lactic acid emerging during hydrolytic degradation. At the initial stage of the degradation course, samples with CaCO_3_ stored in PBS generated higher pH-values than specimens stored in water. After a degradation time of about 200 days, the pH-value of samples with 30–50% CaCO_3_ stored in PBS dropped down below the pH-value of water-stored samples. The reason might be the dissolution of CaCO_3_ particles and the formation of calcium phosphate due to the presence of phosphate ions in PBS solution.

The weight loss of fCM with CaCO_3_ is continuously over time (Figure 5). After 84 days specimens without CaCO_3_ stored in water lost their shape and became paste-like. Trapped water and lactic acid within the specimens were responsible for the short weight increase at this time point. These specimens fully degraded after 220 days of in vitro degradation. The weight loss of fCM with CaCO_3_ depended on the CaCO_3_ content. The higher the CaCO_3_ content, the lower was the weight loss of the fCM specimens.

### 2.5. In Vitro Cytotoxicity Testing of Composite Foams

The in vitro cytotoxicity of the fCM was investigated by both live/dead (FDA/GelRed^®^) staining and WST-1^®^ assay of elution medium using MC3T3-E1 cells.

#### 2.5.1. Live/Dead Staining

Live/dead staining experiments have been performed with composite foams containing 60% (*w*/*w*) of each CaCO_3_ sample batches 47, 02, and SP01. After an incubation period of 1 and 3 days respectively, a large number of viable and only a few dead cells (less than 5%) were found on the composite surfaces. In Figure 6 results of live/dead staining experiments for composite foams containing 60% of different CaCO_3_ sample batches are exemplarily shown.

#### 2.5.2. WST-1^®^ Assay

As shown from Figure 7a, during the first 24 h of elution of the specimens in medium, strong cytotoxicity was observed indicated by a decrease of cellular dehydrogenase activity to about 40% compared to the negative control. Then, until day 14 the relative dehydrogenase activity of MC3T3-E1 cells was in the range between 90 and 100% of the negative control implying a cytocompatible behaviour of the eluates. For the third eluate (day 2–3) and the fifth eluate (day 7–14) the test on the significance of the distance to the negative control even failed substantiating a missing or a very low level of cytotoxicity. The cytotoxicity during the first 24 h is probably caused by the release of DABCO, which was added as foaming catalyst.

In a following experiment the release of cytotoxic components was tested from pre-eluted foams with varying CaCO_3_ concentrations from 0% to 60% (*w*/*w*). These foamed specimens were stored for 72 h in water at 37 °C with a daily change of the water. After drying, the samples were eluted for 24 h with medium as described for the WST-1^®^ assay. The results are shown in Figure 7b. The release of toxic components from specimens with CaCO_3_ was much smaller compared to the first eluate of the unleached specimens shown in Figure 7a, indicated by relative dehydrogenase activities of about 90% of the negative control. Only the foam specimen without CaCO_3_ again showed cytotoxicity.

Concerning this latter specimen without CaCO_3_ addition, different regimes of pre-elution were tested and it could be found that an elution with ethanol for 24 h followed by a subsequent elution for 48 h with water enhanced the cytocompatibility in the WST-1^®^ assay to the same level as found for the CaCO_3_-containing composite samples (results not shown).

### 2.6. Pre-Clinical Pilot Trial

The fCM was evaluated in a model of vertical augmentation in an established animal model on the pigs’ calvaria and on the lateral mandible [54,55,56,57,58]. The use of the established large animal model allows a transfer of the results to the clinical situation [56]. The aim of the animal experiment was to investigate the in vivo behaviour of the material for the first time and to evaluate the in vivo degradation and osteogenic potential regarding vertical augmentation of the pigs’ calotte and the mandible.

The dimensionally stable bone substitute material was easy to handle during the surgical intervention. The operation was performed on all experimental animals without complications. All animals survived the surgical procedure and the following investigation period.

### 2.7. Microradiographic Evaluation

With increasing healing time, there was an increasing resorption of the biomaterial with increasing bone formation. This could be determined both for the augmentation of the skull (Figure 8a) and the mandible (Figure 8c).

The evaluation of the samples on the calvaria showed an increasing amount of new bone formation over time (Figure 8b), while the vertical fragmentation of the lower jaw showed higher new bone formation after 1 month and lower rates of bone formation rates after 2 and 6 months compared to the calvaria (Figure 8d). Interestingly, there were differences in material degradation and bone regeneration both in localization (especially animal A1, A2, A4, A6 and A7) and in interindividual (A3 vs. A4 and A6+A7 vs. A5+A8).

30 Days Postoperative

A large proportion of the bone substitute material was still present. In both animals A1 and A2 bone regeneration predominated in the mandible vs. the skull.

60 Days Postoperative

There were differences between the two animals. While the resorption of the material at the calotte and the lower jaw progressed slowly (except for region 8) and accordingly only little bone regeneration could be observed in test animal 3, high resorption and bone regeneration was seen in test animal 4.

180 Days Postoperative

Most of the fCM has been resorbed (Figure 8a,c). New bone formation was most pronounced in test animal 6, especially on the skull.

### 2.8. Histologic Evaluation

30 Days Postoperative

The fCM was still largely intact in the histological section Figure 9a. Bone ingrowth into the material was not detected at this early stage. Besides the fCM, a pronounced new bone formation was observed. If the block was resorbed in further course, it had already created optimal conditions for new bone formation in the augmented area, since the new bone formation around the block has created a multi-walled defect which showed an optimal regeneration potential.

A similar behaviour was found in the area of the lower jaw. While the fCM was also largely intact here, new bone formation was already visible above the biomaterial. This was caused by the periosteal elevation through the fCM and was also visible on the side opposite the augmentation Figure 9b [55].

60 Days Postoperative

A complete resorption of the fCM was already visible in the area of the skull. The newly formed bone appeared in the fine trabecular structure above the local bone of the skull Figure 9c. The borders of the former cortex were only difficult to detect. Such a result of vertical augmentation can be considered optimal and can otherwise only be achieved with autogenous bone grafting. Another optimal result was achieved in the lower jaw Figure 9d. Here the cortical structure of the mandible was still clearly visible. The augmented area showed complete resorption of the fCM and new bone formation in the sense of lateral augmentation.

After 60 days, however, there were also samples that showed no bone regeneration in the areas after complete resorption of the fCM Figure 9e. New bone was formed in the area of the underside of the mandible and on the opposite side, which was most likely caused by the elongation of the periosteum due to augmentation [55].

180 Days Postoperative

The boundary between augmented area and local bone on the skull was no longer distinguishable Figure 9f. The fCM was completely resorbed and the augmented area completely filled with newly formed bone.

As shown in Figure 9g, in this sample, the new bone formation even clearly exceeded the augmented area. The newly formed bone exceeded the augmented area by more than 5 mm. A neocortical structure was formed in the cranial area. In the cranial area, the former boundary of the augmented area was still visible, while the newly formed bone was already completely fused with the lower jaw in caudal direction Figure 9h.

A further sample showed an almost complete resorption of the fCM with the creation of a bowl-shaped defect in the former augmented area Figure 9i. In the marginal area of the defect a bone regeneration could be seen from the marginal area. Such a defect shows a good regeneration potential with the aim of a vertical augmentation after healing of the newly formed defect.

## 3. Discussion

It was the intention of our work to develop a foamed, i.e., porous, rapidly biodegradable bone substitute material usable for non-load bearing applications in oral, maxillofacial or hand and foot surgery. Our approach based on a comb-like co-oligoester structure synthesized by reaction of a polyol starter (meso-erythritol) with two different lactones (l-lactide, ε-caprolactone). Termination of the free hydroxyl groups of the co-oligoester with lysine ester-derived isocyanate functions yielded the reactive prepolymer.

A structurally similar two-component prepolymer system based on both an isocyanate terminated pentaerythritol and a pentaerythritol tetra(glycolate) with terminal hydroxy groups was described by the group of *Stevens* [43] and processed into polymeric films usable for minimally invasive techniques of bone surgery. Such films showed excellent mechanical properties superior to most of the common bone cements. These results might be the result of the formation of a highly cross-linked polymer network. By addition of 10 wt% of β-calcium triphosphate (β-TCP), non-porous composite films with rougher film surface compared to the unfilled polymers could be prepared. While the mechanical properties were only slightly enhanced by the addition of β-TCP, the β-TCP- containing composite films showed an improved osteoblast cell viability within a 7-days period. Unfortunately, no results are presented about the degradation behavior of these materials.

In our own work, after addition of precipitated CaCO_3_ spherulites as degradable mineral filler material and further reaction accelerating and cross-linkable substances (DMSO, DABCO, LDI, water) foaming of the composite material was initiated and at the same time crosslinking of oligomeric bonds take place. The porosity was achieved by reaction of the isocyanate terminated prepolymer with water resulting in the formation of instable carbamic acid releasing carbon dioxide during decomposition. Porosity of the bone substitute material is intended to enhance attachment and proliferation of bone forming cells after implantation. The pore size depends on the prepolymer/filler ratio of the composite material. By increasing the filler amount, the pores of the resulting foams decrease due to the rising density (Figure 1). The amount and the particle size of the filler also have an influence on the mechanical properties. For our investigations, we have chosen calcium carbonate as mineral filler material by several reasons. It is a rapidly biodegradable material and known for its excellent cytocompatibility. Besides, it is also a source of readily available calcium ions contributing to bone regeneration process. That is why several biomaterials containing CaCO_3_ are already in clinical use. A further important reason is the buffering effect of CaCO_3_ on the emerging lactic acid. Three spherical CaCO_3_ modifications with varying diameter (5, 12 and 22 µm) were incorporated in amounts of 30–60% (*w*/*w*) into the composite foams to vary the mechanical properties of the resulting products. The mechanical properties of fabricated cylindrical specimens were determined by compression test. The specimens were deformed to 50% of their initial height and the resulting compressive strength and modulus were measured. The compressive strength increased with raising CaCO_3_ content and with decreasing particle sizes. Moreover, it was found that the addition of small amounts of DMSO, acting as a softener, has a huge influence on the compressive strength and modulus of the composites. Exemplary, at specimens with CaCO_3_ (SP01, 60% (*w*/*w*)) and 40 µL DMSO per gram prepolymer a compressive strength of 1.7 MPa and a modulus of 4.6 MPa were measured whereas at similar specimens without DMSO addition show a compressive strength of 19.4 MPa and a modulus of 16.7 MPa.

In case of implantation, a rapid decrease in the compressive strength of the fCM in a moist tissue environment may have a negative effect on the mechanical implant stability. When stored in PBS, the investigated fCM show a rapid decline in mechanical stability, which, however, can be remarkably improved by increasing the CaCO_3_ content in composite material (Figure 2). This is also supported by the finding that under in vitro conditions the unfilled polymer material is already completely degraded after approx. 119 days, whereas the fCM with added CaCO_3_ only begins to degrade after 154 days. In vitro degradation experiments of polymeric samples without CaCO_3_ performed both in water and PBS medium exhibited a faster decomposition in water due to the low pH-value of the storage solution, which accelerate the degradation catalytically.

The decreasing level of open porosity with increasing CaCO_3_ content in the fCM composites as shown in Figure 1 may pose a certain problem in ensuring good penetration of the scaffold materials by cells. In contrast to this result, the mechanical stability of the composites during hydrolytic in vitro degradation, especially in the initial phase, with high CaCO_3_ content decreases much less. It therefore seemed necessary to find a workable compromise. In this study, the lower open porosity was chosen in favor of higher mechanical stability, although this problem must be investigated in more detail in further investigations.

When comparing the material with available bone replacement materials, a review by Troeltzsch et al. showed that clinically an augmentation height of 3.7 ± 1.4 mm can be achieved with a particulate bone replacement material [59]. In their work they found that only autogenous bone blocks from extraoral donor sites significantly improve the vertical dimension, while allogeneic and xenogeneic bone block grafts generated significantly less bone height and remained well below 5 mm height gain [59]. In horizontal dimension they found an overall weighted mean gain of 4.5 ± 1.2 mm for allogeneic and xenogeneic block materials, while being associated with high complication rates for allogeneic and xenogeneic materials [59]. Our own working group was able to show preclinically that bone blocks of xenogeneic origin showed significantly less bone formation within the blocks compared to autogenous blocks and therefore did not yet represent a substitute for the autogenous bone graft for vertical augmentation of up to 10–12 mm [58,60,61]. In a retrospective study by Kloss et al. the mean horizontal gain after augmentation was 5.6 ± 1.5 mm for autogenous grafts and 5.5 ± 1.3 mm for allogenic blocks [62]. According to a recent review by Starch-Jensen et al., the work of Kloss et al. is the only work that compares allogeneic blocks with autologous grafts in terms of horizontal augmentation. In non-comparative studies he found a high degree of complications in the use of allogeneic block transplants ranging from dehiscence to complete loss of the graft [63]. In a systematic review, Urban et al. showed that with methods of guided bone regeneration and bone blocks, clinical augmentation heights of between 4.18 and 3.46 mm can be achieved [64].

To the best of our knowledge a composite system comprising CaCO_3_ and a lysine-based urethane monomer has not been described in detail before. There are several studies published on CaCO_3_ containing biodegradable composites with degradable polyesters like polylactide, its copolymers or polycaprolactone for applications in bone regeneration [46,47,48,52,65]. For example, the advantageous buffering capacity of CaCO_3_ was recently confirmed in degradation experiments of poly(DL-lactide)-CaCO_3_ composites in aqueous medium where the pH value remained constant around pH 7 [66]. Only a few reports on CaCO_3_ filled composites with, mostly conventional, polyurethanes have been appeared up to now. A porous composite containing poly (ε-caprolactone) urethane prepared from 4,4′-dicyclohexylmethane diisocyanate and CaCO_3_ (aragonite) as filler was foamed to degradable scaffolds by a salt leaching process. These scaffolds were tested in an in vitro study as scaffolds in bone regeneration showing an improved mechanical stability compared to the unfilled material and a promising bioactivity by the formation of carbonate hydroxyapatite on the scaffold surface [67]. In another paper a lysine-based polyurethane system containing an allograft (bovine bone) component is proposed as an injectable and settable bone biocomposite [68]. This biocomposite supported cellular infiltration and remodelling in femoral condyle defects in rabbits, and there was no evidence of an adverse inflammatory response observed.

Considering that there is currently no suitable material available for vertical and horizontal augmentation heights of up to 10 mm, other than autogenous bone from extraoral donor sites, the pilot study can be considered as a successful evaluation of the fCM block shaped material. The newly developed biomaterial is very easy to process. Due to its high stability and screwability, the material is easy to use clinically. A point that needs to be further investigated in future studies is the rapid decrease in the mechanical strength of the composites within the initial phase of hydrolytic degradation. In the process, the material decomposes quickly and the augmented volume is in many cases filled with bone. Even if the fCM is not directly built through by bone cells, new bone is formed around the fCM due to the material’s biocompatibility of the through periosteal elevation. So that after consecutive resorption of the material, a situation that is biologically difficult to regenerate is how a vertical or lateral augmentation transforms into a multi-walled defect that shows good regeneration potential over time. It should be noted, however, that the results of the in vivo part only result from a small pilot study with a limited number of cases. By combining vertical augmentation on the cranial calotte and lateral augmentation on the mandible, a sample number of 80 could be obtained with only 8 animals at 3 evaluation times.

The promising material has to be tested on a larger number of cases. It shows great potential to replace the autogenous bone graft with this substitute material in selected cases.

## 4. Materials and Methods

### 4.1. Materials

l-Lactide was purchased from Corbion (Gorinchem, The Netherlands). 2,6-Diisocyanato ethyl caproate (l-lysine diisocyanate ethyl ester, LDI) (Infine Chemicals, Shanghai, China) and ε-caprolactone (Sigma-Aldrich, Taufkirchen, Germany) were purified by vacuum distillation. Dichloromethane, chloroform, acetone, heptane and cyclohexane were purchased from Fisher Scientific (Schwerte, Germany), meso-erythritol (Acros, Geel, Belgium), 1,4-Diazabicyclo[2.2.2]octane (DABCO, Sigma-Aldrich, Germany), DMSO (Carl Roth, Karlsruhe, Germany) and stannous octoate (ABCR, Karlsruhe, Germany) were used as received. Phosphate buffered saline tablets (Sigma-Aldrich, Taufkirchen, Germany) were dissolved in water, filtered and sterilized before use.

### 4.2. Prepolymer Synthesis

The isocyanate-terminated oligo(l-lactide-*co*-ε-caprolactone) prepolymer **1b** was synthesized in a two-step process by preparing the oligo(l-lactide-*co*-ε-caprolactone) **1a** followed by endcapping of the formed terminal hydroxyl groups with isocyanate functionalities according to a previously described procedure [30]. In brief, for the first step meso-erythritol (5.0 g, 0.041 mol), l-lactide (59.0 g, 0.41 mol) and ε-caprolactone (18.69 g, 0.164 mol) were reacted with addition of stannous octoate (0.124 g, 0.31 mmol). In the second step LDI was added in a tenfold excess to **1a**. Finally, dichloromethane (50 mL) was added to the reaction mixture and the crude product was isolated by precipitating in cyclohexane and separated by decanting. After repeating this process, the product was dried in vacuum. The prepolymer **1b** was obtained as a colorless viscous oil in a yield of 91%.

### 4.3. Calcium Carbonate Spherulites

Specially designed precipitated calcium carbonates (CaCO_3_, calcite PCC) with almost monodisperse spherical structure of an average particle size of 5 (sample SP01), 12 (sample 47), and 22 µm (sample 02), respectively, provided by SCHAEFER KALK, were used as filler materials in foaming experiments. The SEM micrographs of the CaCO_3_ samples are shown in Figure 10.

### 4.4. Formation of foamed Composite Materials (fCM)

An optimal foaming reaction was obtained by the following procedure: 1 g of **1b** was thoroughly mixed with 30–60 % *w*/*w* calcium carbonate (SP01), 44.6 µL of LDI and 80 µL of DMSO followed by addition of 80 µL of 1,4-diazabicyclo[2.2.2]octane (DABCO; 2.7 M in water). The PU-CaCO_3_ composite paste was filled in cavities of prefabricated silicone moulds (15 mm in diameter, 10 mm in height). The foams were left in the mould for 1 h at room temperature for hardening. The top of the foam was cut off by a scalpel for soft samples and by a microtome (Leica RM 2265, Leica Mikrosysteme, Wetzlar, Germany) for stronger samples, respectively.

### 4.5. Material Characterisation

#### 4.5.1. Polymer Analytics

^1^H-NMR spectroscopy was used to characterize the chemical structures and compositions of the synthesized copolymers **1a** and **1b**. The spectra were recorded using a Varian Inova 500 MHz spectrometer (Varian, Palo Alto, CA, USA) at room temperature using tetramethylsilane as an internal reference and CDCl_3_ as solvent.

**1a**: δ (ppm) = 1.66–1.40 (70.4 H, m, CH_3_ (lactide), 3 x CH_2_ (caprolactone)), 2.41–2.31 (8 H, m, 1 x CH_2_ (caprolactone), 4.36–4.06 (14 H, m, 1 x CH_2_ (caprolactone), 2 x CH_2_ (meso-erythritol, 2 x CH (meso-erythritol)), 5.31–5.05 (14 H, m, CH (lactide)).

**1b**: δ (ppm) = 1.75–1.26 (116.5 H, m, CH_3_ (lactide), 3 x CH_2_ (caprolactone), 3 x CH_2_ (LDI), CH_3_ (LDI)), 2.43–2.29 (8 H, m, 1 x CH_2_ (caprolactone)), 3.36–3.18 (9.7 H, m, 1 x CH2 (LDI)), 4.30–4.02 (24.1 H, m, 1 x CH_2_ (caprolactone), 2 x CH_2_ (meso-erythritol, 2 x CH (meso-erythritol), 1 x CH_2_ (LDI)), 5.19–5.10 (19,1 H, m, CH (lactide), CH (LDI)).

Gel Permeation Chromatography (GPC) was used to determine molecular weights as number-average molecular weight (M_n_) and as weight-average molecular weight (M_w_) with respect to polystyrene standards (PSS-Polymer Standards Service, Mainz, Germany). The degree of dispersion (formerly known as polydispersity index PDI) was determined by the equation Đ = M_w_/M_n_. The measurements were performed on a Shimadzu system (Shimadzu, Duisburg, Germany) equipped with a refraction detector RID 10A (Shimadzu). The samples were dissolved in chloroform at a concentration of 4 mg/mL, and chloroform (stabilised with 1% amylene) was used as eluent, delivered at a flow rate of 1.0 mL min^−1^. As pre-column a PSS-SDV (100 Å, 8.0 × 50 mm) and as column a set of PSS-SDV (100 Å, 8.0 × 300 mm), PSS-SDV (1000 Å, 8,0 × 300 mm) and PSS-SDV (100,000 Å, 8.0 × 300 mm) were used. A refraction detector RID 10A (Shimadzu) was utilized.

#### 4.5.2. Scanning Electron Microscopy (SEM)

Samples (one sample per composition) were examined by a SUPRA 55 VP scanning electron microscope (Carl Zeiss NTS, Oberkochen, Germany). Gold was sputtered onto samples to ensure sufficient electrical conductivity. The images were taken using an InLens-detector with 5 keV excitation energy.

#### 4.5.3. Density and Porosity

The density and open as well as closed cell volume of foamed sample (3 sample replicates per composition) were determined with a pycnometer Ultrapyc 1200e (Quantachrome Instruments, Leipzig, Germany) with nitrogen as process gas.

#### 4.5.4. Mechanical Properties

Compressive strengths of fCM were determined with a texture analyzer TA-XT2i (Stable Micro Systems, Godalming, UK) with a 50 and a 500 N measuring head, respectively. An Inspekt 50 table (Hegewald & Peschke, Nossen, Germany) with a 50 kN measuring head was used for samples where compressive strength was higher than 500 N. For each measurement 3 samples were used. For compression tests the cylindrical samples (15 mm diameter, 9 mm height) were compressed up to 50% at a speed of 0.2 mm/s. Young’s modulus was determined from the slope of the initial linear region of each stress-strain-curve. Foams were measured in dry state and after storage in phosphate buffer saline (PBS) at 37 °C for distinct time intervals.

#### 4.5.5. Hydrolytic Degradation

Studying the in vitro degradation of the fCM the weight loss over time was determined gravimetrically from 3 test specimens of about 300 mg each which were placed in PBS and water, respectively, at 37 °C. After regular time intervals (7 days) medium was removed after pH measurement. Every 28 days the weight change was determined after washing the samples and drying for 48 h at 40 °C. After determination of weight loss, the specimens were stored in medium again. Weight loss was monitored gravimetrically and calculated using the following equation:Weight loss (%) W_i_ = [(W_0_ − W_t_)/W_0_] ×100 % (1)
where W_0_ is the weight of freshly prepared specimen and W_t_ is the weight of dried specimen after storage time t.

#### 4.5.6. In Vitro Cytotoxicity Assay

In vitro cytotoxicity was determined by live/dead and WST-1^®^ assay using cylindrical fCM samples (diameter of 15.3 mm, height of 9 mm). For the live/dead assay the samples (four sample replicates) were disinfected with 70% ethanol (*v*/*v*), rinsed twice with PBS and stored in cell culture medium. The medium was discarded and samples were seeded and cultured with MC3T3-E1 cells (25,000 cells/cm^2^). The nutrient medium was changed on day 2. After 1 and 3 days two of the sample replicates were withdrawn, rinsed with PBS and stained with fluorescein diacetate (FDA, Fluka, Taufkirchen, Germany) and GelRed^®^ (VWR International, Darmstadt, Germany). Living cells deacetylate FDA to green fluorescent fluorescein. GelRed^®^ is able to permeate damaged cell membranes and stains the nuclei of dead cells red fluorescent. The staining procedure allows the estimation of cell density and percentage of dead cells.

To determine extractable cytotoxic components, the foams (three sample replicates) were eluted by incubation in DMEM at 37 °C. According to the international standard ISO 10993-5 a volume of 1 mL of medium was applied per 0.2 g of material. Eluate samples were drawn after 1, 2, 3, 7 and 14 days and the medium was changed. MC3T3-E1 cells with about 8000 cells per well have been incubated with the eluates for 24 h at 37 °C under 5% CO_2_ atmosphere in a 96 well plate (four technical replicates from each eluate sample leading to 12 replicates representing the sample type at a certain time point). Pure cell control medium applied to the cells instead of eluate defined the negative control (NC). Wells without cells defined the positive control (PC). Subsequently, samples and controls were changed against medium with WST-1^®^ reagent (Roche Diagnostics, Mannheim, Germany) and cultured for 1 h. WST-1^®^ conversion by intracellular enzymes was measured as increase of the optical density at 450 nm. Relative dehydrogenase activities were calculated from the raw data by subtraction of the blank value (identical with PC) in the first step and relating all blank corrected values to NC, the metabolic rate of uninfluenced cells, in a second step. The 12 replicates representing a sample type were averaged and differences between the means of the groups were statistically evaluated.

### 4.6. Animal Trial

For the preclinical testing of the novel material, we have decided to conduct a pilot study on the general suitability of the material. The biocompatibility of cylindrical fCM samples (diameter 13.75 mm, height 9 mm) was examined in vivo by an established animal model for bone regeneration [54,55,56,57,58].

#### 4.6.1. Ethical Statement

The study was performed in cooperation with the Semmelweis-University, Budapest, Hungary. The animal husbandry and all animal experiments were carried out in the European Animal Research Centre (“EARC”; 2053 Herceghalom, Hungary, Gesztenyes ut 1; Certified for “Biological evaluation of medical devices” (EN ISO 10993-2:2006)). Specimen preparation, histologic and microradiographic evaluation were performed at the research laboratories of the Department of Oral and Maxillofacial Surgery, University Hospital of Erlangen-Nürnberg (Erlangen, Germany).

#### 4.6.2. Preparation of Biomaterial Specimens.

80 fCM specimens based on CaCO_3_ (60% *w*/*w*, SP01) were used in block form, the edges were removed by milling and a borehole was placed in the middle of each specimen to allow fixation of the samples with an osteosynthesis screw (KLS Martin, Gebrüder Martin; Tuttlingen, Germany). Specimens were sterilized by gamma irradiation.

#### 4.6.3. Animals

Eight female domestic pigs (Sus scrofa domestica) with a bodyweight of 82 ± 3 kg were included in the study. Before the animal experiments were carried out, the animals were kept at a circadian day and night rhythm at a comfortable temperature of 18 ± 1 °C for the animals. The experiments were carried out after a familiarisation phase of 2 weeks. The animals were kept under veterinary supervision during the entire experimental phase.

#### 4.6.4. Anaesthesia Protocol

Prior to surgery, the animals fasted overnight and were then treated according to the following anaesthesia protocol. The intravenous administration of ketamine HCl (Ketavet^®^; Ratiopharm, Ulm, Germany) was followed by an intramuscular injection of medetomidine (Domitor^®^, Pfizer, Karlsruhe, Germany). Perioperative antibiosis was administered 1 h preoperatively and for two days postoperatively to reduce the risk of infection (streptomycin, 0.5 g/day, Grünenthal, Stolberg, Germany). A veterinarian performed anaesthesia and permanent perioperative monitoring of vital parameters. For postoperative pain relief, Temgesic (Temgesic^®^, Böhringer Mannheim, Mannheim, Germany) (0.05 mg/kg every 12 h) was administered for three days after surgery.

#### 4.6.5. Surgical Procedure

First, the animals were placed on their stomachs and a local anaesthetic was applied to the area of the frontal skull (Ultracain DS forte^®^, Sanofi-Aventis Deutschland, Frankfurt am Main, Germany). Subsequently, a sagittal incision was made down to the bone and the soft tissue and periosteum were mobilized. To improve blood flow from the local bone and allow a relatively gap-free application of the fCM samples, approx. 1–2 mm of cortical bone was removed from the skull bone. Four fCM samples per animal were applied and fixed centrally with an osteosynthesis screw (KLS Martin, Gebrüder Martin, Tuttlingen, Germany, (Figure 11, left side of the figure shows a photo of the intraoperative situation, schematic drawing on the left side of the upper right picture).

The periosteum and the skin were then sutured in several layers (Vicryl^®^ 3.0; Vicryl^®^ 1.0; Ethicon, Norderstedt, Germany). Subsequently, the animals were turned on their backs under padding of the operated skull and local anaesthetic was applied on both sides in the area of the lower jaw margin. Now an incision was made in the area of the lower jaw margin on both sides, protecting the facial artery. After a subperiosteal preparation and ablation of the cortical bone analogous to the skull, each three specimens were screwed onto the mandible (Figure 2, at the bottom right of the picture a photo of the intraoperative situation, schematic drawing on the right side of the upper right picture), resulting in a total of 10 specimens for each animal. The periosteum and the skin were then sutured in several layers (Vicryl^®^ 3.0; Vicryl^®^ 1.0).

#### 4.6.6. Sacrifice of Animals and Harvesting of the Specimens

After a healing period of 30 and 60 days, resp., each two and after 180 days four randomly selected animals were sacrificed. The animals were sedated by intramuscular injection of azaperone (1 mg/kg) and midazolam (1 mg/kg). Euthanasia was performed by intravascular injection of 20% pentobarbital solution into an ear vein until cardiac arrest. The skulls were immediately dissected and stored at −80 °C.

#### 4.6.7. Specimen Preparation

A cone beam computed tomography (KaVo 3D eXam, Kavo Dental, Biberach/Riss, Germany) was used to locate the specimens before further preparation. The specimens were then dissected and fixed by immersion in 1.4% paraformaldehyde at room temperature. Specimens were dehydrated through an ascending alcohol series using a tissue processor (Shandon Citadel 1000, Thermo Shandon, Frankfurt, Germany). Xylene was used as an intermediate fixing agent and Technovit 9100 (Heraeus Kulzer, Wertheim, Germany) for embedding. The polymerization was performed in a cold atmosphere (4 °C) to avoid a negative influence of the polymerization heat. After 20 h, the specimens were completely polymerized and cut through their longitudinal axis in the middle of the graded bone blocks with a precision saw (EXAKT Advanced Technologies, Norderstedt, Germany).

### 4.7. Microradiographies

Thin sections were created with a precision saw (EXAKT Advanced Technologies, Norderstedt, Germany) and further reduced to 100–120 µm and polished with a special grinding machine (EXAKT Advanced Technologies, Norderstedt, Germany). With the desktop X-ray unit (Faxitron Cabinet X-ray Systems, Faxitron X-ray, Illinois, IL, USA) at X-ray voltage of 14 kV and 0.25 mA for 2.5 min exposure time, microradiographic images were taken on tooth films of 3.1 x 4.1 cm (Kodak Insight, Rochester, NY, USA). After their development, the microradiographies were digitized with a scanner in grayscale (16 bit, 4800 dpi) (Epson Perfection 4990 Photo, Seiko Epson, Seoul, Korea) and stored as uncompressed files in TIFF format. The images were then analysed with Bioquant OSTEO^®^ software (BIOQUANT Image Analysis, Nashville, TN, USA). A region of interest (ROI) of 1 × 1 cm was defined for evaluation.

### 4.8. Histological Investigation

The specimens were further ground to 25–30 µm thin sections with the special grinding machine. The slides were then transferred in 10% H_2_O_2_ (*w*/*v*) solution for 12 min. After rinsing under running cold water, the specimens were stained with Toluidine Blue O (Sigma-Aldrich Chemie, Munich, Germany) for 12 min. Excess stain was removed by rinsing the specimens under running water. The Toluidine Blue O stained specimens were then digitized with a flatbed scanner (Epson Perfection 4990 Photo, Seiko Epson, Seoul, Korea) as an uncompressed TIFF file with a resolution of 9600 dpi and 16 bit. A region of interest (ROI) of 1 × 1 cm was defined for evaluation.

### 4.9. Statistical Analyses

For the WST-1 assay the differences in the mean values of the sample groups were proven on statistical significance (*p* < 0.05) by one factorial ANOVA. The Holm Sidak test was performed as post hoc test in pairwise comparisons (overall significance level 0.05) using SigmaPlot 11.2 statistical software (Systat Software, San Jose, CA, USA). The Holm Sidak test was recommended by the software as the first line procedure for most multiple comparison testing being more powerful than the Tukey and Bonferroni Tests.

## 5. Conclusion

A composite material composed of biodegradable polyester-urethane and precipitated calcium carbonate spherulites (up to 60% *w*/*w*) was developed and characterised in vitro and in vivo. The fabricated composite material showed excellent biocompatibility and is therefore most suitable for application as bone substitution material. In the preclinical experiment the material showed a simple clinical application and with limited significance of a small pilot study very promising results regarding degradation and bone formation in vertical and lateral augmentation.

## Data Availability

The data presented in this study are available on request from the corresponding author.

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
