# Peer review of "A Novel Resorbable Composite Material Containing Poly(ester-co-urethane) and Precipitated Calcium Carbonate Spherulites for Bone Augmentation—Development and Preclinical Pilot Trials"

_molecules, 2020, doi:10.3390/molecules26010102_

Round 1

Reviewer 1 Report

The reviewed work describes the results of research on the formation and application trials in bone surgery of a porous composite formed with biodegradable polyurethane and pure calcium carbonate. The work is interesting, mainly due to its comprehensive nature and good scientific standard. The authors present the course of the synthesis of segmented polyurethane obtained during the condensation of oligo (L-lactide-co -e-caprolactone) and L-lysine diisocyanate in the presence of calcium carbonate as filler (different types of spherulytic calcite with different grain sizes).

The work is written professionally and may be of interest to a wide range of readers, specialists in the synthesis of biocompatible materials for medicine, as well as experimental surgeons involved in surgery and regeneration of damaged bones therapy. The introduction comprehensively describes the issues related to the subject of the manuscript, although this part should be slightly supplemented and corrected.

The experimental part describing the course of synthesis as well as the characteristics still has to be refined. The part describing cytotoxicity tests and trial on an animal model are carefully made with correct conclusions confirming the initially usefulness of the developed biodegradable composite.

The work should be accepted for publication, but requires some additions.

Introduction.

I believe the authors should be briefly mention about the current use of classical polyurethanes as the imitation bone material for mechanical testing (ASTM F-1839-08 "Standard Specification for Rigid Polyurethane Foam for Use as a Standard Material for Testing Orthopedic Devices and Instruments").

Please discuss other previous research results describing the use of polyurethane composites in bone surgery, e.g. Synthesis of two component injectable polyurethane for bone tissue engineering 2007Biomaterials 28 (3): 423-33

Part Materials and Method

- Please describe in detail the procedure used to measure the porosity of the final composite;

- how long and under what conditions were the samples conditioned before the measurement? The seasoning time and its conditions may have a large influence on the tested later properties of the samples , especially those obtained with the addition of DMSO;

- how were the average molecular weights and the degree of dispersion of the masses of the obtained oligomers determined?

- whether and how was the content of hydroxyl groups in the obtained oligomers determined? This measurement is especially important for the subsequent reaction of this polyol with isocyanates.

Part Results

- there is lack NMR spectra with adequate description, illustrating the actual structure of the obtained oligomers;

- Please correct the scheme 1 of the reaction, it is currently illegible;

- line 342 - authors' note - please delete;

- Please explain why such a rapid decrease in mechanical properties is observed after just 2 hours conditioning in a PBS bath? Is the reason behind the very fast hydrolytic degradation of urethane ester bonds? Are you sure that the complete conversion of the isocyanate groups has been achieved ?  Are there no products of the diisocyanate reaction with water - (can be the water-soluble urea derivatives)?

Dissscusion

Please mention the problem related to, on the one hand, the very long degradation time observed in in vitro conditions, and on the other hand, the relatively rapid decrease in the mechanical properties of this composite. Is this not an obstacle to the proper application of this material as bone active filling?

Author Response

Point-by-point response to the reviewer’ comments

First of all, I would like to express my thanks for the praising and extremely helpful comments of the reviewers. With no doubts these comments will support to substantially improve the scientific content of our manuscript. We now want to go point-by-point through the comments (in Italics) and try to answer questions and propose corrections or additions.

Reviewer 1:

I believe the authors should be briefly mention about the current use of classical polyurethanes as the imitation bone material for mechanical testing (ASTM F-1839-08 "Standard Specification for Rigid Polyurethane Foam for Use as a Standard Material for Testing Orthopedic Devices and Instruments").

Thank you for this interesting advice that we will include in the Introduction.

Please discuss other previous research results describing the use of polyurethane composites in bone surgery, e.g. Synthesis of two component injectable polyurethane for bone tissue engineering 2007Biomaterials 28 (3): 423-33

According to the reviewers comment three further papers describing polyurethane composites including the one mentioned by the reviewer are cited in the introduction. The mentioned paper on injectable polyurethane is discussed more in detail.

- Please describe in detail the procedure used to measure the porosity of the final composite;

As mentioned in the manuscript, the density and open as well as closed cell volume of foamed samples were determined with a pycnometer Ultrapyc 1200e (Quantachrome Instruments) with nitrogen as process gas. Such pycnometers are identified as the instruments of choice to accurately measure the true density of solid materials by employing Archimedes’ principle of fluid displacement, and Boyle’s Law of gas expansion. An inert gas, rather than a liquid (as employed in mercury intrusion-based methods), is used since it will penetrate even the finest pores and eliminate any influence of surface chemistry. The principle of operation is as follows: A sealed sample chamber of known volume is pressurized to a target pressure with the displacement gas. Once stabilized, this pressure is recorded. A valve is then opened allowing the gas to expand into a reference chamber whose volume is also known. Once stabilized, this second pressure is recorded. The pressure drop is then compared to the behavior of the system when a known volume standard underwent the same process. The used method is also recommended to be used for measurements according to ASTM D6226 (Standard Test Method for Open Cell Content of Rigid Cellular Plastics).

In addition, we have recorded Scanning electron microscope images of different batches of the fcm composite material from different positions of the fCM material to illustrate the pore size and inter-connectivity of pores. Results of both methods were in good agreement.

- how long and under what conditions were the samples conditioned before the measurement? The seasoning time and its conditions may have a large influence on the tested later properties of the samples, especially those obtained with the addition of DMSO;

Samples for measurement of porosity were stored for three hours at 23 ± 1 °C under dry conditions (desiccator). For the measurement of mechanical properties of medium-stored fcm samples, the samples were cooled down from 37 °C (measurement temperature) to 23 ± 1 °C, freed from the medium by blotting with filter paper, and measured immediately.

- how were the average molecular weights and the degree of dispersion of the masses of the obtained oligomers determined?

In the manuscript was a subchapter included named polymer analytics. In this new chapter (2.5.1. Polymer analytics), the determination of molecular weights as well as the degree of dispersion by Gel permeation chromatography are described in detail.

- whether and how was the content of hydroxyl groups in the obtained oligomers determined? This measurement is especially important for the subsequent reaction of this polyol with isocyanates.

We agree with the reviewer that the content of hydroxyl groups is important for the subsequent reaction of the polyol with the isocyanate. There are various methods for the determination of the hydroxyl content with regard to urethane formation like e. g. the acetylation of the polyol with an acetylation mixture and the subsequent back titration of the remaining acetic acid with KOH. Procedures for the determination of

so called hydroxyl number are documented e. g. in ASTM E 1899-08 and DIN 53240-2. We have not used this procedure. Based on the structural confirmation of the prepared polyol and its measured molecular weight which was very close to the molecular weight calculated on the basis of the used molar meso-erythritol : L-lactide : ε-caprolactone ratio we assumed the illustrated structure of Scheme 1 with four free terminal hydroxyl groups per prepolymer molecule. In addition, we have used a large excess of LDI isocyanate for the synthesis of prepolymer 1b to be sure to transform most of the hydroxyl groups.

Another important parameter is the isocyanate content (ICC) which can be determined by treatment of the isocyanate derivative with butylamine and back titration of the remaining amine with HCl according to the German standard DIN EN ISO 11909. ICC determination of the isocyanate containing prepolymer resulted in a value of about 7.5 % (mole per cent) related to the entire prepolymer, which is slightly above of the theoretical value (further purification of the prepolymer is needed).

- there is lack NMR spectra with adequate description, illustrating the actual structure of the obtained oligomers;

As mentioned above we have included into the manuscript a new subchapter (2.5.1. Polymer analytics containing now also the NMR data. Results are now discussed in Chapter 3.1. Prepolymer synthesis and characterisation.

- Please correct the scheme 1 of the reaction, it is currently illegible

Sorry for that, we have the position of the scheme now corrected.

- line 342 - authors' note - please delete;

The note was deleted.

- Please explain why such a rapid decrease in mechanical properties is observed after just 2 hours conditioning in a PBS bath? Is the reason behind the very fast hydrolytic degradation of urethane ester bonds? Are you sure that the complete conversion of the isocyanate groups has been achieved ?  Are there no products of the diisocyanate reaction with water - (can be the water-soluble urea derivatives)?

The rapid decrease in mechanical properties within the first hours in aqueous medium is not fully clear for us and needs further investigation in the future. From our point of view there might be several reasons for this behavior. At first, to accelerate the foaming reaction to a medically acceptable level of time, 1,4-diazabicyclo[2.2.2]octane (DABCO) was added to accelerate the foaming reaction. DABCO is a water-soluble, basic, and strongly nucleophilic tertiary amine and might also be able to form adducts with other organic and inorganic derivatives. Although formed fCM samples are carefully washed after foaming, there might be residual DABCO remained in the composite structures releasing during the first hours of the degradation studies. It might be assumed then that DABCO also accelerate the ester cleavage of the oligoester and maybe also the urethane binding structures and thereby reduces the mechanical properties. A second reason might be the relatively weak crosslinking density of the used isocyanate terminated oligomer 1b. In another study to synthesize crosslinked networks from isocyanate-containing oligomers based on different PLA-co-PCL oligoesters we have used Lysine-triisocyanate (2-isocyanatoethyl-2,6-diisocyanato caproylate) to terminate the oligomers. With these derivatives resulting in polymer networks with higher crosslinking degrees are and also improved mechanical properties in the initial phase of hydrolytic degradation are accessible. However, these networks also require a longer time until complete degradation. Furthermore, such triisocyanates tend to undergo spontaneous polymerization reaction. An incomplete conversion of the isocyanate groups can be excluded from our point of view, because we could not detect any free isocyanate groups using different analytical techniques like FT-IR, 13C-NMR, Pyrolysis-GC.

Please mention the problem related to, on the one hand, the very long degradation time observed in in vitro conditions, and on the other hand, the relatively rapid decrease in the mechanical properties of this composite. Is this not an obstacle to the proper application of this material as bone active filling?

Taking into account the current degradation properties of the composite material, there is of course an obstacle when using the fCM in load-bearing areas of the bone. The problem will be mentioned in the discussion. However, with regard to the high porosity of the composite material, this was not the major aim of our study.

The in vivo pilot study confirmed, except for the initial phase, a steady, continuous degradation behavior. More importantly, the study demonstrated, compared to other common artificial substitute materials, the formation of large proportions of newly formed bone tissue around the still existing composite, but degrading material. This result together with the good clinical usability confirm the great potential of the fCM composite as bone substitution material. It should be noted, however, that the results of this in vivo study only result from a small first pilot study with a limited number of animals. The problem is mentioned in the manuscript. In future work a larger number of cases has to be included into an In vivo study to confirm and verify the regenerative potential of this composite material in vertical or lateral bone augmentation.

Reviewer 2 Report

The paper titled “A novel resorbable composite material containing poly(ester-co-urethane) and precipitated calcium carbonate spherulites for bone augmentation development and preclinical pilot trial” appears as extremely interesting after a preliminary reading. This opinion is reinforced after a more profound study of the article was performed.

Nevertheless, this reviewer opines that the chemistry of the article is the less important part of it (besides being highly conventional) and it is the medical field wherein the article is focused. So, in the opinion of this reviewer, the article would be redirected to a more appropriate journal in the biomedical area.

With the independence of the latter, and in the case that the editor considers that a mainly focused on surgery paper may be well accommodated in this journal (I disagree with that), just to say that the article is of great quality.

Nevertheless, the use of Polyurethanes for medical purposes, although is far from being a novelty, have the ability to provide excellent biocompatibility, high and interconnecting porosity, and a certain degree of controlled degradability: By taking in mind these advantages, the authors have prepared an isocyanate-terminated co-oligoester prepolymer with precipitated calcium carbonated spherulites to be used as a bone spare replacer for further dentistry applications. The study itself is very complete, covering all the steps from the synthesis of the polymer to the final applications in vivo, with very good results by using model animals.

The introduction section is very complete, and the very different steps of the experimental work are superb explained in an easy reading style.

This reviewer has enjoyed very much of the work and would like to recommend accepting the paper once the only minor concern that this reviewer has been able to detect concerning the mechanical testing conditions used to determine the compression behavior of the material. Since the compression strength is strongly dependent on the testing rate, please, provide it. This reviewer is sure that due to the high level of all the other experimental descriptions, the absence of the testing rate obeys to a simply forgiving. Please, include this parameter.

With the independence of the mentioned before, and in the opinion of this reviewer the article must be redirected to a more appropriate journal related to any novelties for surgery. Nevertheless, in the case that the editor opines that MOLECULES is the adequate media (I disagree), the paper only requires minor corrections.

Author Response

Point-by-point response to the reviewer’ comments

First of all, I would like to express my thanks for the praising and extremely helpful comments of the reviewers. With no doubts these comments will support to substantially improve the scientific content of our manuscript. We now want to go point-by-point through the comments (in Italics) and try to answer questions and propose corrections or additions.

Reviewer 2:

This reviewer has enjoyed very much of the work and would like to recommend accepting the paper once the only minor concern that this reviewer has been able to detect concerning the mechanical testing conditions used to determine the compression behavior of the material. Since the compression strength is strongly dependent on the testing rate, please, provide it.

Thank for this advice. We have measured the compression properties according to the German DIN EN ISO 3386-1 using a Tensile/compression testing machine, so called texture analyzer with 50 and 500 N measuring heads, respectively, and an Inspekt 50 table with a 50 kN measuring head for samples with a compressive strength higher than 500 N. The used testing rate was 0.2 mm/s. The testing rate was added into the experimental part of the manuscript.

With the independence of the mentioned before, and in the opinion of this reviewer the article must be redirected to a more appropriate journal related to any novelties for surgery.

We consider your comment a great compliment and thank you very much for it. One reason we did not submit the manuscript to a surgery journal is the lack of a comprehensive animal study with an appropriate number of animals to allow statistically significant validation of the results. Within the framework of our investigations, we have only carried out a first pilot study with the newly developed material, which had not been tested under in vivo conditions before. Although the study was extremely positive, we initially did not want to take the risk of sacrificing a large number of animals and to bear the corresponding costs in view of a success that could not be assessed with complete certainty.

Now, based on this successful pilot study and further planned development work to improve the material, we will also be able to publish future results in a higher-ranked surgery journal with a clear conscience. Again, thank you very much.

Round 2

Reviewer 2 Report

In the previous revissión draft this reviewer just suggests minor corrections concerning the contents of the paper. Since, they have been solved by including the testing rate suggested, this reviewer considers that the concern has been solved.

The main criticisms were related to the fact that this reviewer suggested redirecting the paper to a surgery based journal. Since the authors have provided a convincing reason for choosing Molecules, based on the fact that the study is concerned with a pilot study to see the response of the material rather than the clinical aspects requiring more animals to be concluding since a clinical viewpoint,  this reviewer understands these reasons, and so, the previous recommendations have evolved as to suggest accepting the manuscript in its actual state.